# The Effects of Long-Term Magnesium Creatine Chelate Supplementation on Repeated Sprint Ability (RAST) in Elite Soccer Players

**DOI:** 10.3390/nu12102961

**Published:** 2020-09-28

**Authors:** Adam Zajac, Artur Golas, Jakub Chycki, Mateusz Halz, Małgorzata Magdalena Michalczyk

**Affiliations:** Institute of Sport Sciences, The Jerzy Kukuczka Academy of Physical Education in Katowice, Mikolowska 72a, 40-065 Katowice, Poland; a.zajac@awfkatowice.pl (A.Z.); a.golas@awfkatowice.pl (A.G.); j.chycki@awf.katowice.pl (J.C.); mateuszhalz@gmail.com (M.H.)

**Keywords:** creatine, RAST, soccer, anaerobic performance

## Abstract

Aim: The aim of the study was to evaluate the effects of 16 weeks of a low dose of magnesium creatine chelate supplementation on repeated sprint ability test (RAST) results in elite soccer players. Materials: Twenty well-trained soccer players participated in the study. The players were divided randomly into two groups: the supplemented group (SG = 10) and placebo group (PG = 10). Out of the 20 subjects selected for the study, 16 (SG = 8, PG = 8) completed the entire experiment. The SG ingested a single dose of 5500 mg of magnesium creatine chelate (MgCr-C), in 4 capsules per day, which was 0.07 g/kg/d. The PG received an identical 4 capsules containing corn starch. Before and after the study, the RAST was performed. In the RAST, total time (TT), first and sixth 35 m sprint length (s), average power (AP) and max power (MP) were measured. Additionally, before and after the test, lactate LA (mmol/L) and acid–base equilibrium pH (-log(H^+^)), bicarbonates HCO_3_^−^ (mmol/L) were evaluated. Also, in serum at rest, creatinine (mg/dL) concentration was measured. Results: After the study, significantly better results in TT, AP and MP were observed in the SG. No significant changes in the RAST results were observed in the PG. After the study, significant changes in the first 35 m sprint, as well as the sixth 35 m sprint results were registered in the SG, while insignificant changes occurred in the PG. A significantly higher creatinine concentration was observed. Also, a higher post-RAST concentration of LA, HCO_3_^−^ and lower values of pH were observed in April, May and June compared with baseline values. Conclusions: The long timeframe, i.e., 16 weeks, of the low dose of magnesium creatine chelate supplementation improved the RAST results in the SG. Despite the long period of MgCr-C supplementation, in the end of the study, the creatinine level in the SG reached higher but still reference values.

## 1. Introduction

Creatine (Cr) is one of the most often used supplements by competitive athletes [1,2,3]. Numerous studies have shown that oral creatine supplementation with doses of 20–25 g/d for 5–7 days followed by a maintenance dose of 4–5 g/g for several weeks increases intramuscular phosphocreatine (PCr) concentration by stimulating phosphagen metabolism [2,3,4,5]. These metabolic and training adaptations may explain the significant improvements in performance during high-intensity continuous exercise, as in prolonged sprints (400 m dash), or short lasting (5–10 m) repeated bouts of exercise, such as in team sport games or combat sports [6,7,8]. Metabolically, PCr serves as an important energy contributor during high-intensity exercise. Theoretically, it seems that increasing the availability of PCr should enhance power output maintenance during high-intensity exercise and promote recovery between such efforts. Improved performance through Cr supplementation may be related to increased energy availability, as well as enhanced buffering capacity [4]. This is accomplished through the utilization of H^+^ during the creatine kinase CK reaction, and through rephosphorylation of adenosine diphosphorate (ADP) into adenosine triphosphate ATP [9]. Cr supplementation has also been shown to increase the rate of glycogen replenishment, which may help athletes perform prolonged submaximal efforts or repeated high-intensity exercise, requiring both aerobic and anaerobic metabolism [9,10].

There are many sport disciplines which involve a combination of high-intensity actions, in isolation or repeatedly, where optimal anaerobic and aerobic metabolism are required [11]. Soccer is among the sporting activities which are characterized by high-intensity actions, such as sprinting, jumping, kicking, tackling and changes of direction, interspersed by low-intensity jogging or walking [11,12,13,14]. Considering that at the elite level, players cover from 8 to 13 km during a game, the high level of aerobic fitness in soccer seems fully justified [15,16,17]. However, keeping in mind that players perform up to 250 high-intensity actions during a game, it is currently accepted that the decisive actions are covered by anaerobic metabolism [18]. Most of the energy required to resynthesize ATP during such intense short-term efforts are provided by a combination of PCr degradation and anaerobic glycolysis [19]. The proportions between these two metabolic pathways depend on the frequency of high-intensity efforts and the rest intervals between them [12].

Over the last decades, there have been numerous studies evaluating the effects of Cr supplementation on physical performance in soccer, which give conflicting results [1,10,12,17]. Some authors confirm a positive influence of Cr supplementation [1,12,17], yet in most cases, only minor benefits were recorded [20,21,22]. Equivocal results of Cr supplementation have been observed when single action maximal performance was evaluated, such as in short sprints or jumping tests [23]. However, most studies that have involved repeated high-intensity efforts, such as multiple sprints [24,25] or multiple Wingate tests [1,12], showed significant benefits of Cr supplementation. The range of improvement in these studies has varied tremendously, and is most likely related to the dosing and training protocols, as well as the time of the intervention. The majority of available data comes from typical Cr supplementation regimens, which include a loading phase of 20–25 g/d for 5–7 days followed by a maintenance phase of 3–5 g/d for several weeks [2,3]. The minimum duration of Cr supplementation necessary to promote ergogenic effects when ingesting lower doses (0.03 g/kg/d) has not been established, however a recent study [25] has shown that ingesting low doses of Cr (2.3 g/d) for 6 weeks significantly increased plasma Cr levels and improved performance in a multiple high-intensity exercise bout. Also, other factors, such as the form and type of creatine, the method of administration, bioavailability and the type of diet, have a significant influence on the effects of creatine supplementation [26,27]. Considering that each phase of a soccer season lasts several months, few studies have evaluated the effects of Cr supplementation over a period longer than 2 weeks.

To date, there have been no studies on the impact of long-term creatine supplementation on speed endurance performance in male soccer players. Only one study has examined the effects of 13 weeks of creatine supplementation on anaerobic capacity, however the participants were female soccer players [28]. It thus seems fully justified to evaluate the benefits of Cr intake on repeated sprint ability in elite soccer players during 16 weeks of the competitive period. Considering that supplementation with high doses of creatine ethyl esters has been linked to elevated serum creatinine concentration, which is often used as a measure of renal function, the authors decided to monitor this metabolite of creatine throughout the 16-week intervention [29,30,31]. It has been suggested that the creatine loading phase, in which 20–25 g/d is ingested, may significantly increase serum creatinine concentration up to 2–2.5 mmol/L and decrease the glomerular filtration rate (GFR) to 25–35 mL/d [32].

The main objective of this study was to evaluate the long-term effects of Cr supplementation on repeated sprint abilities, crucial for soccer without the classical loading phase, which often causes gastrointestinal problems or muscle cramping. We incorporated the intervention period into the 16 week second round soccer competition to obtain information about fitness variables, especially speed endurance, speed as well as changes in body mass and chosen biochemical variables related to buffering capacity.

## 2. Materials and Methods

### 2.1. Subjects

Twenty well-trained elite soccer players participated in the study. The players were divided randomly into two groups: the supplemented group (SG = 10) and the placebo group (PG = 10). From the 20 subjects selected for the study, 16 (3 forwards, 5 midfielders, 6 defenders, 2 goalkeepers) completed the entire experiment. Two athletes dropped out of the research due to injury while one resolved his contract and one did not finish the study due to illness. The SG (*n* = 8) received creatine chelate magnesium and the placebo group (*n* = 8) received identical capsules filled with starch. The experiment took place during the second part of the season, from the beginning of March until the end of June and lasted 16 weeks. The athletes constituted a homogenous group in regard to age. The participants’ somatic characteristics are presented in Table 1. The athletes were well trained, as the study period was preceded by eight weeks of general and specific conditioning. All subjects had valid medical examinations and showed no contraindications to participate in the study. The players were informed verbally and in writing about the experimental protocol, the possible risks and benefits of the study and the possibility to withdraw at any stage of the experiment. All participants gave their written consent for participation. The study was approved by the Research Ethics Committee at the Academy of Physical Education in Katowice, Poland (KB-7/2018).

### 2.2. Diet and Supplementation Protocol

Energy as well as macro- and micro-nutrient intake of all subjects were determined by the 72 h nutrition recall (two weekdays and one weekend day), 2 weeks before the study was initiated [8]. The participants were placed on an isocaloric (3455 ± 436 kcal/day) mixed diet (55% carbohydrates, 20% protein, 25% fat) prior to and during the investigation. The main meals were prepared in the club’s cafeteria. The athletes were allowed to drink water as desired (2.5–3 L/day). The pre-trial meals were consumed 2.5 h before the anaerobic sprint test. The meals were standardized for energy intake (600 kcal) and consisted of carbohydrates (80%), fats (10%) and protein (10%). The participants did not take any medications and substances not prescribed by the supplementation protocol for 4 weeks before and during the study. The subjects in the SG ingested a single dose of 5500 mg of magnesium creatine chelate (MgCr-C), in 4 capsules per day, which was 0.07 g/kg/d, while drinking a high glycemic fruit juice 10–15 min after daily practice sessions. The control group ingested identical capsules containing corn starch.

### 2.3. Study Protocol

The subjects underwent medical examinations and somatic measurements in the morning, between 08:00 and 08:30. Body composition was evaluated by the electrical impedance method (Inbody 720, Biospace Co., Anaheim, Los Angeles, CA, USA). The day before these evaluations, the participants had their last meal at 20:00 and were allowed to drink 1.5 L of water between 21:00 and 23:00 to fully hydrate the body. They reported to the laboratory after an overnight fast, refraining from exercise for 24 h. The measurements of body mass were performed on a medical scale with a precision of 0.1 kg. The experiment lasted 16 weeks, during which the Repeated-Anaerobic Sprint Test was performed 5 times, which included baseline evaluations and four post-intervention trials following creatine supplementation or placebo intake. The participants refrained from exercise for one day before testing to minimize the effects of fatigue.

During the study period, which included the second part of the competitive season, the athletes trained on an everyday basis (approximately 2 h/d) with an official soccer game on Saturday/Sunday. Additionally, twice a week, the players performed a strength and conditioning training session.

### 2.4. Anaerobic Performance Test

Anaerobic performance was evaluated by the Running Anaerobic Sprint Test (RAST) protocol which involved 6 × 35 m maximal sprint efforts, separated by 10 s of active recovery [33]. Infrared photocell gates (Witty, Micro Gate System, Mahopac, NY, USA) were placed precisely 35 m apart. The photocell system was used to evaluate the sprint times at 35 m. The 35 m distance evaluated absolute speed, while total time of the 6 × 35 m sprints determined the level of speed endurance and anaerobic capacity. Additionally, peak and mean power values were calculated by using the following formula: Power = body mass × distance ÷ time. The RAST was performed on artificial grass in an indoor facility to avoid variance in atmospheric conditions. Temperature was between 18 and 20 degrees, no wind and humidity of 55–60%. The participants were verbally informed about the time of the rest interval between particular sprints. Before testing, the players completed a 15 min warm-up, which included jogging, dynamic stretching as well as several starts and accelerations. After a 5 min passive rest, they reported to the starting line and began the RAST protocol on a command. The subjects were instructed to sprint the 35 m distance as fast as possible, decelerate after the finish line and jog back to the starting line for the next repetition. The procedure was repeated until 6 sprints were completed (Figure 1).

### 2.5. Biochemical Assays

To determine lactate concentration (LA), and acid–base equilibrium, the following variables were evaluated: LA (mmol/L), blood pH (-log(H^+^)) and bicarbonates (HCO_3_^−^ std mmol/L). The evaluations were performed from fingertip capillary blood samples, at rest and after 3 min of post-exercise recovery. Determination of LA was based on an enzymatic method (Biosen C-line Clinic, EKF-diagnostic GmbH, Barleben, Germany). The remaining acid–base variables were assessed using a Blood Gas Analyzer GEM 3500 (Analyzer Premier 3500, GEM, Bedfort, MA, USA). For creatinine analysis, 4.5 mL of venous blood from the antecubital vein was collected and a MAK079 (Sigma Aldrich, St. Louis, MO, USA) kit test was used.

### 2.6. Statistical Analysis

The Shapiro–Wilk, Levene and Mauchly’s tests were used to verify the normality, homogeneity and sphericity of the samples’ data variances, respectively. Verifications of the differences between the considered values before and after creatine supplementation, between baseline conditions and the 4 time points in the SG and PG groups were verified using analysis of variance (ANOVA) with repeated measures. Effect sizes (Cohen’s d) were reported where appropriate. According to Hopkins guidelines, the effect size (eta-squared; η^2^) was established as follows: 0.01—small, 0.06—medium and 0.14—large [34]. Statistical significance was set at *p* < 0.05. All statistical analyses were performed using Statistica 9.1 (TIBCO Software Inc., Palo Alto, CA, USA) and Microsoft Office (Redmont, Washington, DC, USA), and are presented as means with standard deviations (SD).

## 3. Results

The intragroup ANOVA analysis revealed significant differences between baseline conditions and the four time points in LA post-exercise, pH at rest, pH post-exercise, HCO_3_^−^ at rest and HCO_3_^−^ post-exercise in the SG (Table 2). A significant effect and higher values of LA post-exercise and HCO_3_^−^ post-exercise, and lower values of pH post-exercise were observed in April (respectively *p* = 0.002, η^2^ = 0.20; *p* = 0.005, η^2^ = 0.18; *p* = 0.007, η^2^ = 0.14), May (respectively *p* = 0.001, η^2^ = 0.25; *p* = 0.001, η^2^ = 0.27; *p* = 0.001, η^2^ = 0.20) and June (respectively *p* = 0.001, η^2^ = 0.33; *p* = 0.001, η^2^ = 0.30; *p* = 0.001, η^2^ = 0.21) compared to baseline values. Simultaneously, higher values of HCO_3_^−^ at rest and pH at rest in March (respectively *p* = 0.002, η^2^ = 0.18; *p* = 0.002, η^2^ = 0.22), April (respectively *p* = 0.001, η^2^ = 0.23; *p* = 0.001, η^2^ = 0.28), May (respectively *p* = 0.001, η^2^ = 0.28; *p* = 0.001, η^2^ = 0.32) and June (respectively *p* = 0.001, η^2^ = 0.31; *p* = 0.001, η^2^ = 0.35) were observed compared to baseline values. There were no statistically significant intra-group differences between baseline conditions and the four time points of LA concentration at rest in the SG (Table 2).

The intragroup ANOVA analysis in the PG revealed significant differences between baseline conditions and the four time points in LA concentration at rest, LA post-exercise, pH post-exercise, HCO_3_^−^ at rest and HCO_3_^−^ post-exercise. There was no statistically significant intragroup difference between baseline conditions and the four time points of LA concentration at rest and blood pH at rest in the PG (Table 3). A significant effect and higher values of HCO_3_^−^ at rest and HCO_3_^−^ post-exercise, and lower values of pH post-exercise were observed in April (respectively *p* = 0.001, η^2^ = 0.06; *p* = 0.005, η^2^ = 0.05; *p* = 0.003, η^2^ = 0.07), May (respectively *p* = 0.001, η^2^ = 0.09; *p* = 0.001, η^2^ = 0.08; *p* = 0.001, η^2^ = 0.10) and June (respectively *p* = 0.001, η^2^ = 0.11; *p* = 0.001, η^2^ = 0.12; *p* = 0.001, η^2^ = 0.13) compared to baseline values. Higher values of LA concentration post-exercise in March (*p* = 0.02, η^2^ = 0.07), April (*p* = 0.001, η^2^ = 0.09), May (*p* = 0.001, η^2^ = 0.12) and June (*p* = 0.001, η^2^ = 0.18) were observed compared to baseline values in the PG. Serum creatinine levels did not change significantly throughout the 16-week intervention in the PG, and ranged from 0.92 mmol/L at baseline to 0.85 mmol/L during the last evaluation in June. In the SG, the creatinine concentration rose from 0.83 mmol/L at baseline in February to 1.87 mmol/L (*p* = 0.01) in March and decreased in the following time points, reaching 1.38 mmol/L (*p* = 0.05) during the last evaluation in June after 16 weeks of continuous supplementation (Figure 2).

A significant effect of the interventions on the TT, the AP and the MP in the intragroup measures was revealed. A significant decrease of the TT (*p* = 0.001, η^2^ = 0.22), and a significant increase of the AP (*p* = 0.005, η^2^ = 0.11) and the MP (*p* = 0.007, η^2^ = 0.09) were observed in the SG. No significant changes in these variables were observed in the PG (Table 3). A significant effect of the long-term supplementation with MgCr-C on performance in the first 35 m sprint (*p* = 0.05, η^2^ = 0.13), as well as the sixth 35 m (*p* = 0.001, η^2^ = 0.32) sprint was registered in the SG, while insignificant changes occurred in the PG.

The intergroup ANOVA showed lower values of TT after the intervention (*p* = 0.001, η^2^ = 0.36) as well as higher values of MP (*p* = 0.003, η^2^ = 0.11) and AP (*p* = 0.001, η^2^ = 0.18) in the SG compared to the PG (Table 3).

## 4. Discussion

In this study, we evaluated the influence of 16 weeks of magnesium chelate creatine supplementation on RAST results in elite soccer players. Our study is unique on a global scale and its uniqueness consists of three aspects. The first is the duration of the experiment, which lasted 16 weeks. Most previous studies with Cr supplementation lasted on the average from 5 to 7 days, up to a few weeks [17,21,24,35]. Conducting long-term supplementation research with soccer players or other contact sport disciplines is quite difficult, as such sports expose players to injuries, which in turn automatically exclude them from the research [13]. In addition, during the soccer season, players are transferred to other clubs, which makes data collection difficult or impossible. Therefore, often, the number of subjects, as in the present study, decreases during long-term research. Moreover, the coaches are not willing to cooperate in this type of research explaining that it may introduce additional stress for the players, and exercise tests disrupt their training process [13]. The second novel aspect of this study is the use of low doses of Cr supplementation without the typical loading phase, in which 20–25 g/d are ingested over the first week. Previously, scientists studied the effects of creatine monohydrate (CrM), ingested with high doses (20–25 g/d) over 4–5 days, followed by a maintenance dose of 4–5 g for the next few days of the protocol [1,2]. Finally, the third unique aspect of this study was the use of an exercise test validated and recommended specifically for soccer players, the Repeated Anaerobic Sprint Test (RAST), in order to assess changes in anaerobic capacity after Cr supplementation [36]. In previous studies with Cr supplementation in soccer players, anaerobic abilities were assessed by tests which were not specific for this team sport [12,23,37,38].

The most popular Cr supplementation procedure assumes a loading phase for the first few days by administering high doses of Cr (20–25 g/d), followed by a maintenance dose of 4–5 g/d for a few weeks [1,35]. In our study, however, the loading phase was eliminated. For 16 weeks, the subjects in the SG consumed low doses of Cr (4–5 g/d) which amounted to 0.07 g/kg/d. The advantage of skipping the loading phase was the absence of any adverse gastrointestinal incidents, which are frequent and can diminish players’ performance or even exclude them from competition. The longest period of Cr supplementation that has been studied with the participation of soccer players has been 9 weeks [39]. As mentioned before, in the present study, Cr supplementation lasted 16 weeks, during which period the players did not show any adverse renal or urinary symptoms. Compared to protocols with loading (20–25 g of Cr/d), in which serum creatinine concentrations often rise significantly past the reference range (0.6–1.4 mg/dL), our study showed an increase in this metabolite after the first 4 weeks of supplementation (1.87 mg/dL), which declined over the next 3 months, reaching the reference range at the end of the 16-week experiment (1.38 mg/dLl) (Figure 2). This confirms results of previous research that long-term Cr supplementation does not adversely affect kidney function [40]. The supplementation protocol brought numerous physiological benefits to the athletes, such as increased phosphagen content and muscle buffering capacity, which were maintained for 4 months (Table 1). The ergogenic effects were mostly evidenced in the repeated sprint test (RAST), where the SG reached significantly better results than the PG (Table 2). A rarely observed phenomenon in creatine supplementation protocols with competitive athletes is the improvement in speed, which was quite significant in the group ingesting MgCr-C. The first 35 m sprint improved steadily over the 16-week period, from 5.15 s at baseline to 4.91 s in the last evaluation in June. The speed endurance ergogenic effect was even more evidenced when the sixth 35 m sprints were compared at baseline and after each 4-week testing period. The players in the SG improved from 5.93 s in February to 5.43 s in June at the end of the season.

In soccer, it is known that there is a strong positive correlation between game performance and the number of repeated sprints during a game [14,18,36]. Throughout the soccer match, forwards, midfielders and defenders perform sprints mainly at 5 to 10 m distances [18,36]. During sprints, when re-synthesizing ATP, the muscles first use PCr and then muscle glycogen. In the SG group, the RAST results demonstrated a significant difference between baseline and post-intervention values compared with those of the PG. The results were probably the consequence of an increased PCr pool in muscle cells of the lower limbs [41]. This contributed to a greater participation of PCr in ATP re-synthesis in both the first and the last sprint of the RAST (Table 3). On the other hand, in the PG group, where the phosphagen pool did not increase, in the last sprint, ATP re-synthesis probably occurred through glycolysis, which caused a significant decrease in sprint performance. Additionally, in the SG, CrMg-C supplementation increased the buffering capacity, which was evidenced by higher post-exercise LA concentration and higher blood pH level (Table 2). This is the second most significant role of Cr during exercise. Cr increases the intracellular buffering capacity in muscles by binding hydrogen ions released from the breakdown of ATP [9]. The higher LA and HCO_3_^−^ levels observed in the SG, as well as the lower blood pH compared with better RAST results, show that despite the high rate of ATP re-synthesis both from the breakdown of PCr stores and from glycolysis, which promotes cellular acidosis, the soccer players were buffered against acidosis. Despite the low blood pH (Table 2), which also indicates low muscle pH, subjects from the SG were able to achieve significantly better results than before the supplementation as well as when compared with the PG. An efficient buffering mechanism allowed the SG to continue their efforts despite increased acidosis. In the PG, worse RAST results in all stages of the experiment (Table 3) were probably caused by lower PCr muscle concentration, slower ATP re-synthesis through glycolysis and lower muscle buffering capacity. In the present study, in both groups after subsequent sprints, LA blood concentration increased, which suggests that the activity of glycolytic enzymes and thus the ATP re-synthesis was slower. Additionally, the results of increased acidosis were visible through the level of HCO_3_^−^ and pH and also in the results of the first and last sprint. Beside Cr, LA is essential in removing pyruvate, regenerating nicotinamide adenine dinucleotide NAD^+^ to sustain a high rate of ATP regeneration from glycolysis and contributes to metabolic proton buffering [5].

In most of the studies conducted so far, the impact of Cr supplementation on anaerobic performance in soccer players has brought positive results [21,42,43]. Ostojic and Mujika [21] concluded that acute Cr supplementation favorably affected repeated sprint performance and limited the decline in jumping ability in highly trained soccer players. Similar effects of Cr supplementation in soccer players have been presented by other authors [21,25]. Ambiguous results of Cr supplementation in soccer players have been observed, especially when single action maximum performance was evaluated, such as in the Wingate or jumping tests [9,12,23,44]. Using several repetitions of the Wingate test to assess anaerobic fitness of soccer players does not seem appropriate because the test is performed on an ergocycle, and this form of exercise does not reflect the specificity of the effort during soccer competition or training [12]. On the other hand, Cox et al. [25] and Williams et al. [43] tested the effects of Cr supplementation during a soccer match-simulated effort and achieved satisfying results [25,43]. Considering the above, the RAST seems to be the best physical fitness test for soccer players to monitor anaerobic performance during training routines [42]. The validity of the RAST has been investigated in previous studies, and its high reliability and validity have been repeatedly confirmed [45]. Furthermore, a high reliability of the RAST has been confirmed, even when performed on less rigid surfaces such as grass [42]. In this study, in both groups, the RAST was performed five times, at baseline (0′), after 4 (1′), after 8 (2′), after twelve (3′) and after 16 (4′) weeks of MgC-Cr supplementation, regular training and competition. The results (Table 3) revealed that 16 weeks of CrMg-C supplementation improved repeated sprint ability (s), PP (W) and MP (W). Ramirez et al. [42], observed equally significant differences in the repeated sprint ability test in soccer players after 6 days of 5 g/d Cr supplementation. On the other hand, the effect of short acute CrM loading, 0.3 g per kg over 5 days, on single repeated sprints and fatigue index values measured in the RAST were also studied by Ateş et al. [46]. In their study, there were no significant differences in RAST results following the supplementation with CrM.

Other factors which influence the results of Cr supplementation include the daily dosing and the exercise and supplementation protocol implemented in the study [11]. The majority of available data comes from typical Cr supplementation regimens, which include a loading phase for 5–7 days followed by a maintenance phase lasting for 5 to several weeks [11,32]. In the present study, the loading phase was skipped and a low dose, 4–5 g/d, of CrMg-C for 16 weeks was used. The positive effect of such long-term supplementation was evidenced by better RAST results in the SG, which continued throughout the 4 months of training and competition (Table 3). Cr is a polar and hydrophilic molecule due to its composition, which limits Cr bioavailability [47]. CrM, the most popular form of Cr used in anaerobic exercise studies is not resistant to acidic and alkaline solutions [26]. In an alkaline surrounding (high pH), as in the duodenum and initial sections of the small intestine, much of the supplemented Cr is irreversibly converted to the inactive form of creatinine [26]. To maintain Cr stability in conditions of variable pH of the gastrointestinal tract, the best solutions are still being sought for combining Cr with other compounds that are to prevent its irreversible conversion to creatinine [47]. The form of magnesium creatine chelate which was administered in this study is more stable and does not cyclize to such an extent as creatine monohydrate [27]. Another factor which increases Cr bioavailability includes a high-carbohydrate, low-fat diet [27]. Moreover, consuming Cr with carbohydrates or with a carbohydrate-protein supplements increases its intestinal absorption [26]. During the research, the SG consumed a mixed diet which was rich in carbohydrates. Additionally, in order to increase CrMg^+^ absorption, the SG consumed it with a high-carbohydrate fruit juice [26]. So far, few studies have evaluated the effects of Cr supplementation over a period longer than 6 weeks [23,28,39]. Claudino et al. [23], after a 7-week period of CrM supplementation in 14 Brazilian elite soccer players, observed a lack of decline in the strength of the muscles of the lower extremities compared with a control group. The research was conducted in the preparatory period, in which the soccer players consumed CrM over the first week in a dose of 20 g/d and for the next six weeks, they ingested 5 g/d. To the best of our knowledge, only one study investigated the chronic effects of creatine supplementation while training, but it regarded female soccer players [28]. These authors showed that 13 weeks of CrM supplementation (2 × 7.5 g/d in the first week and 5 g/d throughout the rest of the protocol) improved muscle strength but not lean mass in collegiate female soccer players [28].

Creatine is known to cause mild water retention and to decrease urinary volume due to its osmotic effect [4]. The absorption of Cr into a muscle cell entails an increased inflow of water aimed at equalization of osmotic pressure on both sides of the cytoplasmic membrane [4]. The increase in osmotic pressure in the cell results in the activation of mitogen-activated protein kinase (MAPK) stress response proteins, which, among other things, induce myocyte differentiation through satellite cell involvement [47]. In the SG, a significant increase in BM and FFM after the first 4 weeks of CrMg-C confirms this theory. Other researchers also observed similar results [26]. Brilla et al. [27] examined the effects of magnesium-creatine supplementation on body water and quadriceps’ torque. The group supplemented with Mg-creatine chelate displayed significant increases in their intra-cellular water, as well as quadriceps’ peak torque. The lack of further growth of the FFM in the SG is due to the fact that the study was carried out during the spring–summer competitive period and the players were constantly subjected to intense efforts, both during matches and training. Such conditions are not conducive to synthesizing muscle proteins but rather regenerating muscle micro-damage and a continuous muscle degradation process [47].

## 5. Conclusions

The long-term, 16-week, low-dose magnesium creatine chelate supplementation improved RAST results in elite soccer players during the second part of the competitive season. Compared to the placebo group, speed as well as power increased over the intervention period, with small increases in body mass and fat-free mass over the first 4 weeks of supplementation. Despite the 16-week period of MgCr-C supplementation, blood creatinine levels remained within reference values at the end of the season.

Given the effect of creatine on the anaerobic capacity of athletes, research into its various forms over an extended period of time seems justified and may be of great practical significance for team sport athletes, where the competitive season lasts for several months.

## Figures and Tables

**Figure 1 nutrients-12-02961-f001:**
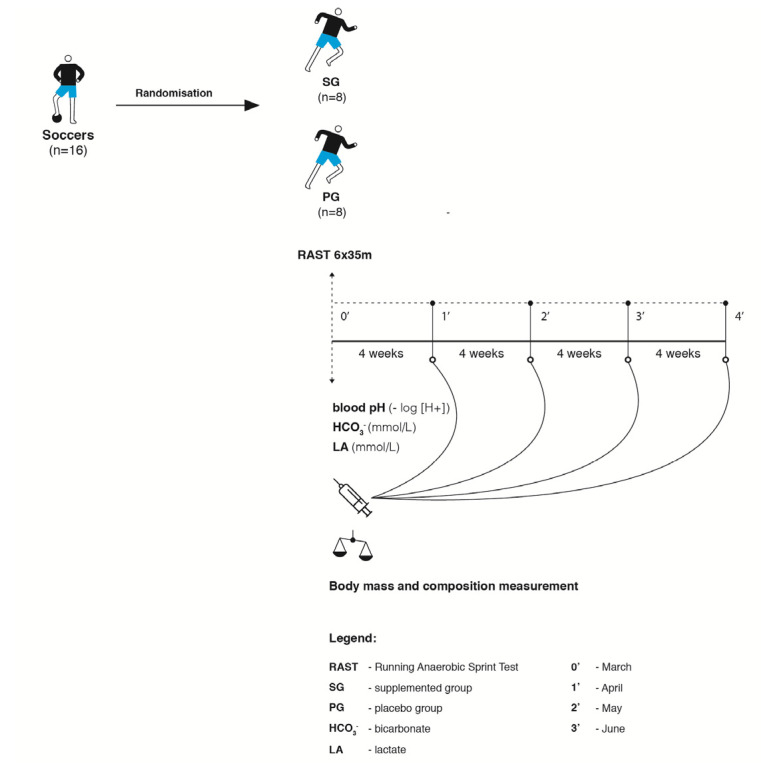
Scheme of testing procedures.

**Figure 2 nutrients-12-02961-f002:**
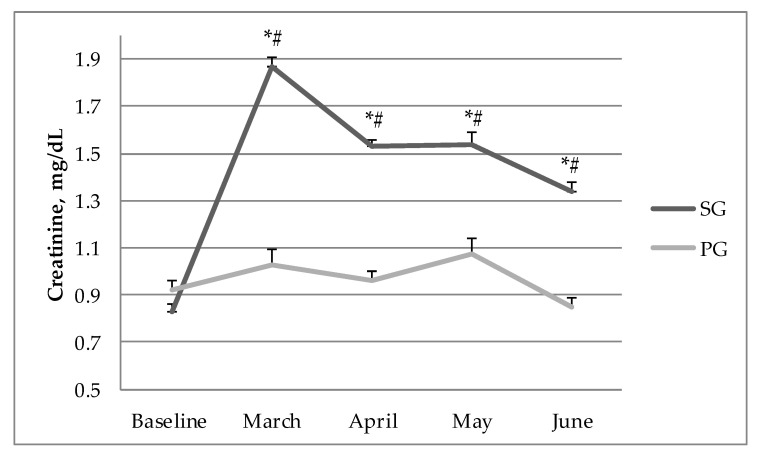
Serum creatinine concentration at baseline and during four months of supplementation in the SG and PG. SG—supplemented group, PG—placebo group. * significant difference compared with baseline in the SG, # significant difference compared with the PG.

**Table 1 nutrients-12-02961-t001:** The participants’ somatic characteristics.

Variables	Values
Age, years	25.6 ± 3.7
Hight, cm	181.2 ± 4.5
Weight, kg	78.3 ± 5.3
Body Fat,%	9.6 ± 2.4

**Table 2 nutrients-12-02961-t002:** Biochemical variables before and after the supplementation protocol.

Variables	SG, Before	SG, After	PG, Before	PG, After
	Mean ± SD	Mean ± SD	Mean ± SD	Mean ± SD
LA rest, mmol/L	1.14 ± 0.21	1.35 ± 0.12	1.16 ± 0.17	1.38 ± 0.12
LA post-exercise mmol/L	7.11 ± 0.33	9.03 ± 0.82 *	6.45 ± 0.37	7.54 ± 0.28
pH rest (-log(H^+^))	7.34 ± 0.01	7.41 ± 0.01 *	7.34 ± 0.02	7.36 ± 0.05
pH post-exercise (-log(H^+^))	7.31 ± 0.03	7.20 ± 0.02 *	7.34 ± 0.02	7.26 ± 0.02
HCO_3_^−^ rest, mmol/L	23.44 ± 0.31	24.83 ± 0.34 *	23.30 ± 0.33	23.93 ± 0.20
HCO_3_^−^ post-exercise, mmol/L	15.11 ± 0.62	17.04 ± 0.60 *	14.64 ± 0.47	15.70 ± 0.49
Creatinine, mg/dL	0.83 ± 0.16	1.34 ± 0.23 *	0.92 ± 0.11	0.85 ± 0.16

Note: SG—supplemented group, PG—placebo group, LA—lactate, HCO_3_^−^—bicarbonate, SD—standard deviation. * significant differences between baseline and post intervention values in the SG and the PG.

**Table 3 nutrients-12-02961-t003:** The Running Anaerobic Sprint Test (RAST) results before and after the supplementation intervention.

Variables	SG, Before	SG, After	PG, Before	PG, After
	Mean ± SD	Mean ± SD	Mean ± SD	Mean ± SD
TT, (s)	33.37 ± 0.54	31.21 ± 0.63 *	33.65 ± 1.13	33.42 ± 0.94
First 35 m sprint length, (s)	5.17 ± 0.14	4.91 ± 0.09 *	5.18 ± 0.14	5.18 ± 0.13
Sixth 35 m sprint length, (s)	5.93 ± 0.13	5.43 ± 0.08 *	5.89 ± 0.23	5.81 ± 0.07
MP, (W)	681.92 ± 87.07	786.74 ± 79.81 *	691.12 ± 88.09	695.27 ± 95.92
AP, (W)	561.28 ± 66.75	678.50 ± 65.77 *	568.50 ± 88.07	577.07 ± 79.99

Note: TT—total time, MP—Max power, AP—average power, SG—supplemented group, PG—placebo group, RAST—Running Anaerobic Sprint Test. * significant differences between the SG and PG after supplementation.

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
