# Peer review of "The Effects of Long-Term Magnesium Creatine Chelate Supplementation on Repeated Sprint Ability (RAST) in Elite Soccer Players"

_nutrients, 2020, doi:10.3390/nu12102961_

Round 1

Reviewer 1 Report

The current manuscript describes a small human study in "elite" soccer players ingesting a magnesium creatine chelate for 16 weeks and measuring anaerobic performance vis the RAST at five different time points. The originality and novelty of the study are not exceptional (as the authors state themselves creatine has been a well known ergogenic aid for power since the 70’s), however, the study population once further defined is a hard to access population which strengthens the results and conclusions. Kudos to the authors there. I have some edits and suggestions that I think would improve the manuscripts. Please see below:

Intro: The information here is good, however, it does need some editing by a native English speaker. VERY minor but just to help with the flow of information. The authors have all the information needed in the intro but I would recommend spending less time talking about the biochemistry of creatine and more about the functional outcomes and the importance it has in sporting activities. This would really help sell the study and why it was done in the first place. The biochemistry of creatine is really interesting stuff to read and I think some of it still needed; i.e., its role in resynthesizing ATP and been a relatively weak buffer but not much more outside of that is needed since this isn’t a review.

Materials and Methods:

Line 110: Define “elite.”

Line 117 & 118: Put the demongraphics in a table that also show them for each group. In addition, show the type of each position player in each group (e.g., 3 forwards, 2 midfielders, etc) breakdown.

Stats: say what type of ANOVA you ran for what variable. Right now I’m not really sure how you ran your state – I have an idea of how you did it but I’m not 100% certain. I would state “A 2X5 [group (SG and PG) X time (week0, week 4, week 8, week 12, week 16)] repeated measures ANOVA was used to assess differences between group at individual time points for *insert variables*. Please state how the data is presented, e.g., mean±standard deviation.

Results: In your tables you have a few spots where there is a “,” and there should be a “.” Please fix these in both tables. Figure 2 should include a variance measure not just the mean.

Discussion: This section has a lot of good information but is entirely too long for an original research manuscript. It reads basically like a literature review. I would restructure it like this: discuss what your results are, how they relates to previous literature, and provide context for why your results were different or similar based on the physiological understanding of creatine’s mechanism of action. I would have any opening paragraph reiterating the results and what was observed in the present study. Then I would discuss the repeated sprints results as that’s the main functional outcome in the paper. Next I would do a paragraph on the blood variables measured. I would then mention any limitations of the current study followed by a paragraph on the practical applications of the study and then finally your conclusion paragraph. All the information is there already I just think this will streamline the discussion, make it shorter, and easier to read for anyone interested in the topic.

Author Response

Reviewer#1

Comments and Suggestions for Authors

The current manuscript describes a small human study in "elite" soccer players ingesting a magnesium creatine chelate for 16 weeks and measuring anaerobic performance vis the RAST at five different time points. The originality and novelty of the study are not exceptional (as the authors state themselves creatine has been a well known ergogenic aid for power since the 70’s), however, the study population once further defined is a hard to access population which strengthens the results and conclusions. Kudos to the authors there. I have some edits and suggestions that I think would improve the manuscripts. Please see below:

Intro: The information here is good, however, it does need some editing by a native English speaker. VERY minor but just to help with the flow of information. The authors have all the information needed in the intro but I would recommend spending less time talking about the biochemistry of creatine and more about the functional outcomes and the importance it has in sporting activities. This would really help sell the study and why it was done in the first place. The biochemistry of creatine is really interesting stuff to read and I think some of it still needed; i.e., its role in resynthesizing ATP and been a relatively weak buffer but not much more outside of that is needed since this isn’t a review.

Answer: Done as requested

Materials and Methods:

Line 110: Define “elite.”

Line 117 & 118: Put the demongraphics in a table that also show them for each group. In addition, show the type of each position player in each group (e.g., 3 forwards, 2 midfielders, etc) breakdown.

Answer: done as request

Stats: say what type of ANOVA you ran for what variable. Right now I’m not really sure how you ran your stats – I have an idea of how you did it but I’m not 100% certain. I would state “A 2X5 [group (SG and PG) X time (week0, week 4, week 8, week 12, week 16)] repeated measures ANOVA was used to assess differences between group at individual time points for *insert variables*. Please state how the data is presented, e.g., mean ± standard deviation.

Answer: Verifications of the differences between the considered values before and after creatine supplementation, between baseline conditions and the 4 time points in the SG and PG groups were verified using ANOVA with repeated measures.

Results: In your tables you have a few spots where there is a “,” and there should be a “.” Please fix these in both tables. Figure 2 should include a variance measure not just the mean.

Answer: done as requested

Discussion: This section has a lot of good information but is entirely too long for an original research manuscript. It reads basically like a literature review. I would restructure it like this: discuss what your results are, how they relates to previous literature, and provide context for why your results were different or similar based on the physiological understanding of creatine’s mechanism of action. I would have any opening paragraph reiterating the results and what was observed in the present study. Then I would discuss the repeated sprints results as that’s the main functional outcome in the paper. Next I would do a paragraph on the blood variables measured. I would then mention any limitations of the current study followed by a paragraph on the practical applications of the study and then finally your conclusion paragraph. All the information is there already I just think this will streamline the discussion, make it shorter, and easier to read for anyone interested in the topic.

Answer: We fully agree with the reviewer that the discussion is lengthy, as we attempted to put too much basic knowledge in that part. According to you suggestions we have concentrated on explaining the mechanisms behind the improved RAST, in the context of long term chronic adaptive changes. The blood variables will follow, yet this part will be significantly limited, just to support the outcomes of the repeated sprints test results following Cr ingestion. The practical implications and limitations of the study are presented next, yet they are also limited to the most relevant facts. The conclusions will be more precise and easier to read for those interested in the topic.

Reviewer#1

Comments and Suggestions for Authors

The current manuscript describes a small human study in "elite" soccer players ingesting a magnesium creatine chelate for 16 weeks and measuring anaerobic performance vis the RAST at five different time points. The originality and novelty of the study are not exceptional (as the authors state themselves creatine has been a well known ergogenic aid for power since the 70’s), however, the study population once further defined is a hard to access population which strengthens the results and conclusions. Kudos to the authors there. I have some edits and suggestions that I think would improve the manuscripts. Please see below:

Intro: The information here is good, however, it does need some editing by a native English speaker. VERY minor but just to help with the flow of information. The authors have all the information needed in the intro but I would recommend spending less time talking about the biochemistry of creatine and more about the functional outcomes and the importance it has in sporting activities. This would really help sell the study and why it was done in the first place. The biochemistry of creatine is really interesting stuff to read and I think some of it still needed; i.e., its role in resynthesizing ATP and been a relatively weak buffer but not much more outside of that is needed since this isn’t a review.

Answer: Done as requested

Materials and Methods:

Line 110: Define “elite.”

Line 117 & 118: Put the demongraphics in a table that also show them for each group. In addition, show the type of each position player in each group (e.g., 3 forwards, 2 midfielders, etc) breakdown.

Answer: done as request

Stats: say what type of ANOVA you ran for what variable. Right now I’m not really sure how you ran your stats – I have an idea of how you did it but I’m not 100% certain. I would state “A 2X5 [group (SG and PG) X time (week0, week 4, week 8, week 12, week 16)] repeated measures ANOVA was used to assess differences between group at individual time points for *insert variables*. Please state how the data is presented, e.g., mean ± standard deviation.

Answer: Verifications of the differences between the considered values before and after creatine supplementation, between baseline conditions and the 4 time points in the SG and PG groups were verified using ANOVA with repeated measures.

Results: In your tables you have a few spots where there is a “,” and there should be a “.” Please fix these in both tables. Figure 2 should include a variance measure not just the mean.

Answer: done as requested

Discussion: This section has a lot of good information but is entirely too long for an original research manuscript. It reads basically like a literature review. I would restructure it like this: discuss what your results are, how they relates to previous literature, and provide context for why your results were different or similar based on the physiological understanding of creatine’s mechanism of action. I would have any opening paragraph reiterating the results and what was observed in the present study. Then I would discuss the repeated sprints results as that’s the main functional outcome in the paper. Next I would do a paragraph on the blood variables measured. I would then mention any limitations of the current study followed by a paragraph on the practical applications of the study and then finally your conclusion paragraph. All the information is there already I just think this will streamline the discussion, make it shorter, and easier to read for anyone interested in the topic.

Answer: We fully agree with the reviewer that the discussion is lengthy, as we attempted to put too much basic knowledge in that part. According to you suggestions we have concentrated on explaining the mechanisms behind the improved RAST, in the context of long term chronic adaptive changes. The blood variables will follow, yet this part will be significantly limited, just to support the outcomes of the repeated sprints test results following Cr ingestion. The practical implications and limitations of the study are presented next, yet they are also limited to the most relevant facts. The conclusions will be more precise and easier to read for those interested in the topic.

Reviewer#1

Comments and Suggestions for Authors

The current manuscript describes a small human study in "elite" soccer players ingesting a magnesium creatine chelate for 16 weeks and measuring anaerobic performance vis the RAST at five different time points. The originality and novelty of the study are not exceptional (as the authors state themselves creatine has been a well known ergogenic aid for power since the 70’s), however, the study population once further defined is a hard to access population which strengthens the results and conclusions. Kudos to the authors there. I have some edits and suggestions that I think would improve the manuscripts. Please see below:

Intro: The information here is good, however, it does need some editing by a native English speaker. VERY minor but just to help with the flow of information. The authors have all the information needed in the intro but I would recommend spending less time talking about the biochemistry of creatine and more about the functional outcomes and the importance it has in sporting activities. This would really help sell the study and why it was done in the first place. The biochemistry of creatine is really interesting stuff to read and I think some of it still needed; i.e., its role in resynthesizing ATP and been a relatively weak buffer but not much more outside of that is needed since this isn’t a review.

Answer: Done as requested

Materials and Methods:

Line 110: Define “elite.”

Line 117 & 118: Put the demongraphics in a table that also show them for each group. In addition, show the type of each position player in each group (e.g., 3 forwards, 2 midfielders, etc) breakdown.

Answer: done as request

Stats: say what type of ANOVA you ran for what variable. Right now I’m not really sure how you ran your stats – I have an idea of how you did it but I’m not 100% certain. I would state “A 2X5 [group (SG and PG) X time (week0, week 4, week 8, week 12, week 16)] repeated measures ANOVA was used to assess differences between group at individual time points for *insert variables*. Please state how the data is presented, e.g., mean ± standard deviation.

Answer: Verifications of the differences between the considered values before and after creatine supplementation, between baseline conditions and the 4 time points in the SG and PG groups were verified using ANOVA with repeated measures.

Results: In your tables you have a few spots where there is a “,” and there should be a “.” Please fix these in both tables. Figure 2 should include a variance measure not just the mean.

Answer: done as requested

Discussion: This section has a lot of good information but is entirely too long for an original research manuscript. It reads basically like a literature review. I would restructure it like this: discuss what your results are, how they relates to previous literature, and provide context for why your results were different or similar based on the physiological understanding of creatine’s mechanism of action. I would have any opening paragraph reiterating the results and what was observed in the present study. Then I would discuss the repeated sprints results as that’s the main functional outcome in the paper. Next I would do a paragraph on the blood variables measured. I would then mention any limitations of the current study followed by a paragraph on the practical applications of the study and then finally your conclusion paragraph. All the information is there already I just think this will streamline the discussion, make it shorter, and easier to read for anyone interested in the topic.

Answer: We fully agree with the reviewer that the discussion is lengthy, as we attempted to put too much basic knowledge in that part. According to you suggestions we have concentrated on explaining the mechanisms behind the improved RAST, in the context of long term chronic adaptive changes. The blood variables will follow, yet this part will be significantly limited, just to support the outcomes of the repeated sprints test results following Cr ingestion. The practical implications and limitations of the study are presented next, yet they are also limited to the most relevant facts. The conclusions will be more precise and easier to read for those interested in the topic.

Reviewer#1

Comments and Suggestions for Authors

The current manuscript describes a small human study in "elite" soccer players ingesting a magnesium creatine chelate for 16 weeks and measuring anaerobic performance vis the RAST at five different time points. The originality and novelty of the study are not exceptional (as the authors state themselves creatine has been a well known ergogenic aid for power since the 70’s), however, the study population once further defined is a hard to access population which strengthens the results and conclusions. Kudos to the authors there. I have some edits and suggestions that I think would improve the manuscripts. Please see below:

Intro: The information here is good, however, it does need some editing by a native English speaker. VERY minor but just to help with the flow of information. The authors have all the information needed in the intro but I would recommend spending less time talking about the biochemistry of creatine and more about the functional outcomes and the importance it has in sporting activities. This would really help sell the study and why it was done in the first place. The biochemistry of creatine is really interesting stuff to read and I think some of it still needed; i.e., its role in resynthesizing ATP and been a relatively weak buffer but not much more outside of that is needed since this isn’t a review.

Answer: Done as requested

Materials and Methods:

Line 110: Define “elite.”

Line 117 & 118: Put the demongraphics in a table that also show them for each group. In addition, show the type of each position player in each group (e.g., 3 forwards, 2 midfielders, etc) breakdown.

Answer: done as request

Stats: say what type of ANOVA you ran for what variable. Right now I’m not really sure how you ran your stats – I have an idea of how you did it but I’m not 100% certain. I would state “A 2X5 [group (SG and PG) X time (week0, week 4, week 8, week 12, week 16)] repeated measures ANOVA was used to assess differences between group at individual time points for *insert variables*. Please state how the data is presented, e.g., mean ± standard deviation.

Answer: Verifications of the differences between the considered values before and after creatine supplementation, between baseline conditions and the 4 time points in the SG and PG groups were verified using ANOVA with repeated measures.

Results: In your tables you have a few spots where there is a “,” and there should be a “.” Please fix these in both tables. Figure 2 should include a variance measure not just the mean.

Answer: done as requested

Discussion: This section has a lot of good information but is entirely too long for an original research manuscript. It reads basically like a literature review. I would restructure it like this: discuss what your results are, how they relates to previous literature, and provide context for why your results were different or similar based on the physiological understanding of creatine’s mechanism of action. I would have any opening paragraph reiterating the results and what was observed in the present study. Then I would discuss the repeated sprints results as that’s the main functional outcome in the paper. Next I would do a paragraph on the blood variables measured. I would then mention any limitations of the current study followed by a paragraph on the practical applications of the study and then finally your conclusion paragraph. All the information is there already I just think this will streamline the discussion, make it shorter, and easier to read for anyone interested in the topic.

Answer: We fully agree with the reviewer that the discussion is lengthy, as we attempted to put too much basic knowledge in that part. According to you suggestions we have concentrated on explaining the mechanisms behind the improved RAST, in the context of long term chronic adaptive changes. The blood variables will follow, yet this part will be significantly limited, just to support the outcomes of the repeated sprints test results following Cr ingestion. The practical implications and limitations of the study are presented next, yet they are also limited to the most relevant facts. The conclusions will be more precise and easier to read for those interested in the topic.

Reviewer#1

Comments and Suggestions for Authors

The current manuscript describes a small human study in "elite" soccer players ingesting a magnesium creatine chelate for 16 weeks and measuring anaerobic performance vis the RAST at five different time points. The originality and novelty of the study are not exceptional (as the authors state themselves creatine has been a well known ergogenic aid for power since the 70’s), however, the study population once further defined is a hard to access population which strengthens the results and conclusions. Kudos to the authors there. I have some edits and suggestions that I think would improve the manuscripts. Please see below:

Intro: The information here is good, however, it does need some editing by a native English speaker. VERY minor but just to help with the flow of information. The authors have all the information needed in the intro but I would recommend spending less time talking about the biochemistry of creatine and more about the functional outcomes and the importance it has in sporting activities. This would really help sell the study and why it was done in the first place. The biochemistry of creatine is really interesting stuff to read and I think some of it still needed; i.e., its role in resynthesizing ATP and been a relatively weak buffer but not much more outside of that is needed since this isn’t a review.

Answer: Done as requested

Materials and Methods:

Line 110: Define “elite.”

Line 117 & 118: Put the demongraphics in a table that also show them for each group. In addition, show the type of each position player in each group (e.g., 3 forwards, 2 midfielders, etc) breakdown.

Answer: done as request

Stats: say what type of ANOVA you ran for what variable. Right now I’m not really sure how you ran your stats – I have an idea of how you did it but I’m not 100% certain. I would state “A 2X5 [group (SG and PG) X time (week0, week 4, week 8, week 12, week 16)] repeated measures ANOVA was used to assess differences between group at individual time points for *insert variables*. Please state how the data is presented, e.g., mean ± standard deviation.

Answer: Verifications of the differences between the considered values before and after creatine supplementation, between baseline conditions and the 4 time points in the SG and PG groups were verified using ANOVA with repeated measures.

Results: In your tables you have a few spots where there is a “,” and there should be a “.” Please fix these in both tables. Figure 2 should include a variance measure not just the mean.

Answer: done as requested

Discussion: This section has a lot of good information but is entirely too long for an original research manuscript. It reads basically like a literature review. I would restructure it like this: discuss what your results are, how they relates to previous literature, and provide context for why your results were different or similar based on the physiological understanding of creatine’s mechanism of action. I would have any opening paragraph reiterating the results and what was observed in the present study. Then I would discuss the repeated sprints results as that’s the main functional outcome in the paper. Next I would do a paragraph on the blood variables measured. I would then mention any limitations of the current study followed by a paragraph on the practical applications of the study and then finally your conclusion paragraph. All the information is there already I just think this will streamline the discussion, make it shorter, and easier to read for anyone interested in the topic.

Answer: We fully agree with the reviewer that the discussion is lengthy, as we attempted to put too much basic knowledge in that part. According to you suggestions we have concentrated on explaining the mechanisms behind the improved RAST, in the context of long term chronic adaptive changes. The blood variables will follow, yet this part will be significantly limited, just to support the outcomes of the repeated sprints test results following Cr ingestion. The practical implications and limitations of the study are presented next, yet they are also limited to the most relevant facts. The conclusions will be more precise and easier to read for those interested in the topic.

Reviewer#1

Comments and Suggestions for Authors

The current manuscript describes a small human study in "elite" soccer players ingesting a magnesium creatine chelate for 16 weeks and measuring anaerobic performance vis the RAST at five different time points. The originality and novelty of the study are not exceptional (as the authors state themselves creatine has been a well known ergogenic aid for power since the 70’s), however, the study population once further defined is a hard to access population which strengthens the results and conclusions. Kudos to the authors there. I have some edits and suggestions that I think would improve the manuscripts. Please see below:

Intro: The information here is good, however, it does need some editing by a native English speaker. VERY minor but just to help with the flow of information. The authors have all the information needed in the intro but I would recommend spending less time talking about the biochemistry of creatine and more about the functional outcomes and the importance it has in sporting activities. This would really help sell the study and why it was done in the first place. The biochemistry of creatine is really interesting stuff to read and I think some of it still needed; i.e., its role in resynthesizing ATP and been a relatively weak buffer but not much more outside of that is needed since this isn’t a review.

Answer: Done as requested

Materials and Methods:

Line 110: Define “elite.”

Line 117 & 118: Put the demongraphics in a table that also show them for each group. In addition, show the type of each position player in each group (e.g., 3 forwards, 2 midfielders, etc) breakdown.

Answer: done as request

Stats: say what type of ANOVA you ran for what variable. Right now I’m not really sure how you ran your stats – I have an idea of how you did it but I’m not 100% certain. I would state “A 2X5 [group (SG and PG) X time (week0, week 4, week 8, week 12, week 16)] repeated measures ANOVA was used to assess differences between group at individual time points for *insert variables*. Please state how the data is presented, e.g., mean ± standard deviation.

Answer: Verifications of the differences between the considered values before and after creatine supplementation, between baseline conditions and the 4 time points in the SG and PG groups were verified using ANOVA with repeated measures.

Results: In your tables you have a few spots where there is a “,” and there should be a “.” Please fix these in both tables. Figure 2 should include a variance measure not just the mean.

Answer: done as requested

Discussion: This section has a lot of good information but is entirely too long for an original research manuscript. It reads basically like a literature review. I would restructure it like this: discuss what your results are, how they relates to previous literature, and provide context for why your results were different or similar based on the physiological understanding of creatine’s mechanism of action. I would have any opening paragraph reiterating the results and what was observed in the present study. Then I would discuss the repeated sprints results as that’s the main functional outcome in the paper. Next I would do a paragraph on the blood variables measured. I would then mention any limitations of the current study followed by a paragraph on the practical applications of the study and then finally your conclusion paragraph. All the information is there already I just think this will streamline the discussion, make it shorter, and easier to read for anyone interested in the topic.

Answer: We fully agree with the reviewer that the discussion is lengthy, as we attempted to put too much basic knowledge in that part. According to you suggestions we have concentrated on explaining the mechanisms behind the improved RAST, in the context of long term chronic adaptive changes. The blood variables will follow, yet this part will be significantly limited, just to support the outcomes of the repeated sprints test results following Cr ingestion. The practical implications and limitations of the study are presented next, yet they are also limited to the most relevant facts. The conclusions will be more precise and easier to read for those interested in the topic.

Reviewer#1

Comments and Suggestions for Authors

The current manuscript describes a small human study in "elite" soccer players ingesting a magnesium creatine chelate for 16 weeks and measuring anaerobic performance vis the RAST at five different time points. The originality and novelty of the study are not exceptional (as the authors state themselves creatine has been a well known ergogenic aid for power since the 70’s), however, the study population once further defined is a hard to access population which strengthens the results and conclusions. Kudos to the authors there. I have some edits and suggestions that I think would improve the manuscripts. Please see below:

Intro: The information here is good, however, it does need some editing by a native English speaker. VERY minor but just to help with the flow of information. The authors have all the information needed in the intro but I would recommend spending less time talking about the biochemistry of creatine and more about the functional outcomes and the importance it has in sporting activities. This would really help sell the study and why it was done in the first place. The biochemistry of creatine is really interesting stuff to read and I think some of it still needed; i.e., its role in resynthesizing ATP and been a relatively weak buffer but not much more outside of that is needed since this isn’t a review.

Answer: Done as requested

Materials and Methods:

Line 110: Define “elite.”

Line 117 & 118: Put the demongraphics in a table that also show them for each group. In addition, show the type of each position player in each group (e.g., 3 forwards, 2 midfielders, etc) breakdown.

Answer: done as request

Stats: say what type of ANOVA you ran for what variable. Right now I’m not really sure how you ran your stats – I have an idea of how you did it but I’m not 100% certain. I would state “A 2X5 [group (SG and PG) X time (week0, week 4, week 8, week 12, week 16)] repeated measures ANOVA was used to assess differences between group at individual time points for *insert variables*. Please state how the data is presented, e.g., mean ± standard deviation.

Answer: Verifications of the differences between the considered values before and after creatine supplementation, between baseline conditions and the 4 time points in the SG and PG groups were verified using ANOVA with repeated measures.

Results: In your tables you have a few spots where there is a “,” and there should be a “.” Please fix these in both tables. Figure 2 should include a variance measure not just the mean.

Answer: done as requested

Discussion: This section has a lot of good information but is entirely too long for an original research manuscript. It reads basically like a literature review. I would restructure it like this: discuss what your results are, how they relates to previous literature, and provide context for why your results were different or similar based on the physiological understanding of creatine’s mechanism of action. I would have any opening paragraph reiterating the results and what was observed in the present study. Then I would discuss the repeated sprints results as that’s the main functional outcome in the paper. Next I would do a paragraph on the blood variables measured. I would then mention any limitations of the current study followed by a paragraph on the practical applications of the study and then finally your conclusion paragraph. All the information is there already I just think this will streamline the discussion, make it shorter, and easier to read for anyone interested in the topic.

Answer: We fully agree with the reviewer that the discussion is lengthy, as we attempted to put too much basic knowledge in that part. According to you suggestions we have concentrated on explaining the mechanisms behind the improved RAST, in the context of long term chronic adaptive changes. The blood variables will follow, yet this part will be significantly limited, just to support the outcomes of the repeated sprints test results following Cr ingestion. The practical implications and limitations of the study are presented next, yet they are also limited to the most relevant facts. The conclusions will be more precise and easier to read for those interested in the topic.

Reviewer#1

Comments and Suggestions for Authors

The current manuscript describes a small human study in "elite" soccer players ingesting a magnesium creatine chelate for 16 weeks and measuring anaerobic performance vis the RAST at five different time points. The originality and novelty of the study are not exceptional (as the authors state themselves creatine has been a well known ergogenic aid for power since the 70’s), however, the study population once further defined is a hard to access population which strengthens the results and conclusions. Kudos to the authors there. I have some edits and suggestions that I think would improve the manuscripts. Please see below:

Intro: The information here is good, however, it does need some editing by a native English speaker. VERY minor but just to help with the flow of information. The authors have all the information needed in the intro but I would recommend spending less time talking about the biochemistry of creatine and more about the functional outcomes and the importance it has in sporting activities. This would really help sell the study and why it was done in the first place. The biochemistry of creatine is really interesting stuff to read and I think some of it still needed; i.e., its role in resynthesizing ATP and been a relatively weak buffer but not much more outside of that is needed since this isn’t a review.

Answer: Done as requested

Materials and Methods:

Line 110: Define “elite.”

Line 117 & 118: Put the demongraphics in a table that also show them for each group. In addition, show the type of each position player in each group (e.g., 3 forwards, 2 midfielders, etc) breakdown.

Answer: done as request

Stats: say what type of ANOVA you ran for what variable. Right now I’m not really sure how you ran your stats – I have an idea of how you did it but I’m not 100% certain. I would state “A 2X5 [group (SG and PG) X time (week0, week 4, week 8, week 12, week 16)] repeated measures ANOVA was used to assess differences between group at individual time points for *insert variables*. Please state how the data is presented, e.g., mean ± standard deviation.

Answer: Verifications of the differences between the considered values before and after creatine supplementation, between baseline conditions and the 4 time points in the SG and PG groups were verified using ANOVA with repeated measures.

Results: In your tables you have a few spots where there is a “,” and there should be a “.” Please fix these in both tables. Figure 2 should include a variance measure not just the mean.

Answer: done as requested

Discussion: This section has a lot of good information but is entirely too long for an original research manuscript. It reads basically like a literature review. I would restructure it like this: discuss what your results are, how they relates to previous literature, and provide context for why your results were different or similar based on the physiological understanding of creatine’s mechanism of action. I would have any opening paragraph reiterating the results and what was observed in the present study. Then I would discuss the repeated sprints results as that’s the main functional outcome in the paper. Next I would do a paragraph on the blood variables measured. I would then mention any limitations of the current study followed by a paragraph on the practical applications of the study and then finally your conclusion paragraph. All the information is there already I just think this will streamline the discussion, make it shorter, and easier to read for anyone interested in the topic.

Answer: We fully agree with the reviewer that the discussion is lengthy, as we attempted to put too much basic knowledge in that part. According to you suggestions we have concentrated on explaining the mechanisms behind the improved RAST, in the context of long term chronic adaptive changes. The blood variables will follow, yet this part will be significantly limited, just to support the outcomes of the repeated sprints test results following Cr ingestion. The practical implications and limitations of the study are presented next, yet they are also limited to the most relevant facts. The conclusions will be more precise and easier to read for those interested in the topic.

Reviewer 2 Report

Introduction

Line 55. We would appreciate, please, that you include the work of Fierro and Urzua, 2013, in your references: Almonacid Fierro, MA and Urzua Alul, LA. The impact of the supply of creatine monohydrate in canoeing athletes. Revista Iberoamericana de Ciencias de la Actividad Física y el Deporte 2013,1(1):1-19.

Line 70. The inclusion of all football participants is interesting. Referees are part of the game and the following study provides relevant information about this research which we encourage you to consider. Rebolé, M., Castillo, D., Cámara, J., and Yanci, J. “Relationship between the cardiovascular capacity and repeated sprints ability in high-standard soccer referees. Revista Iberoamericana de Ciencias de la Actividad Física y el Deporte 2016,5 (3):49-64.

Line 104. Please remove the space before number 37. It should look like this [37,38]

Line 106. Please, change 2 - 2.5mmol/l by 2-2.5mmol/l

The introduction is very well written and reviews in depth the state of the art in relation to what is to be researched. However, the identification of the objectives should be better explained at the end of this section.

It is implicitly suggested that it is this: "It thus seems fully justified to evaluate the benefits of Cr intake on repeated sprint ability in elite soccer players during 16 weeks of the competitive period". But it leaves serious doubts if what you want to check are the effects of Creatine intake during 16 weeks or its control in serum ... or both. My advice is to make the small adjustments in the text that will allow you to identify without doubt the objective(s). ...The work is excellent!

Material and Methods

Line 112. Please include a dot after the parenthesis (PG= 10).

Line 123. Please include the ethics committee approval code.

Line 127. Please include which 24-hour validated test you have used and include the reference.

Line 136. Please remove one of the dots after "sessions”.

Line 141. Please separate the letter "l" from 1.5l

Line 182. I congratulate you and thank you for informing us about the effect-size

Results

Line 241. Please remove one of the dots after "intervention. .”

Line 241. In the last two rows of the table, the comma (,) has been used as a decimal separator. Please replace all commas with dots.

Line 243. Figure 3 > In Figure 3, the variable pH rest does not show data for PG and the variable LA rest does not show values for any of the groups.

This must be solved or explained because it confuses the reader.

Line 245. In Figure 3, as you have done in previous tables and figures, include the description of the abbreviations that appear.

Discussion

Line 436. Please remove the space before number 35. It should look like this [35]

Line 437. The reference number 351 appears. This is an error that should be checked

Line 448. Please remove one of the dots after "training.”

Without a doubt the discussion is the best part of this article, without the previous ones not being good as well. It makes an excellent comparison with the results of previous work which gives this work enormous potential to be revered in the future.

References

Please check each reference one by one and in detail. There are minor formatting errors that require your attention to finally bring you up to speed with a journal like this.

Thanks

Author Response

Reviewer #2

Introduction

Line 55. We would appreciate, please, that you include the work of Fierro and Urzua, 2013, in your references: Almonacid Fierro, MA and Urzua Alul, LA. The impact of the supply of creatine monohydrate in canoeing athletes. Revista Iberoamericana de Ciencias de la Actividad Física y el Deporte 2013,1(1):1-19.

Line 70. The inclusion of all football participants is interesting. Referees are part of the game and the following study provides relevant information about this research which we encourage you to consider. Rebolé, M., Castillo, D., Cámara, J., and Yanci, J. “Relationship between the cardiovascular capacity and repeated sprints ability in high-standard soccer referees. Revista Iberoamericana de Ciencias de la Actividad Física y el Deporte 2016,5 (3):49-64.

Line 104. Please remove the space before number 37. It should look like this [37,38]

Line 106. Please, change 2 - 2.5mmol/l by 2-2.5mmol/l

Answer: All changes were done as requested

The introduction is very well written and reviews in depth the state of the art in relation to what is to be researched. However, the identification of the objectives should be better explained at the end of this section.

Answer: Thank you for your valuable observations but we decided not to shorten the admission. We believe that a detailed explanation of the biochemical processes taking place with the participation of creatine significantly facilitates further understanding of the advisability of supplementation with magnesium creatine chelate. However, if the reviewer, after reading the text in the changed form, again decides that the introduction should be shortened, we will do it.

It is implicitly suggested that it is this: "It thus seems fully justified to evaluate the benefits of Cr intake on repeated sprint ability in elite soccer players during 16 weeks of the competitive period". But it leaves serious doubts if what you want to check are the effects of Creatine intake during 16 weeks or its control in serum ... or both. My advice is to make the small adjustments in the text that will allow you to identify without doubt the objective(s). ...The work is excellent!

The main objective was to evaluate the long term effects of Cr supplementation on repeated sprint abilities, crucial for soccer without the classical loading phase, which often causes gastrointestinal problems or muscle cramping. We incorporated the intervention period into the 16 weeks second round competition to obtain information about fitness variables, predominant speed endurance, speed as well as changes in body mass and chosen biochemical variables related to buffering capacity.

Answer: Thank you for your valuable attention. We have added the above text to the introduction.

Material and Methods

Line 112. Please include a dot after the parenthesis (PG= 10).

Line 123. Please include the ethics committee approval code.

Line 127. Please include which 24-hour validated test you have used and include the reference.

Line 136. Please remove one of the dots after "sessions”.

Line 141. Please separate the letter "l" from 1.5l

Line 182. I congratulate you and thank you for informing us about the effect-size

Answer: All changes were done as requested

Results

Line 241. Please remove one of the dots after "intervention. .”- done

Line 241. In the last two rows of the table, the comma (,) has been used as a decimal separator. Please replace all commas with dots.- done

Line 243. Figure 3 > In Figure 3, the variable pH rest does not show data for PG and the variable LA rest does not show values for any of the groups. This must be solved or explained because it confuses the reader.

Line 245. In Figure 3, as you have done in previous tables and figures, include the description of the abbreviations that appear.

 Answer: Thank you for valuable comments. Finally figure 3 was deleted.

Discussion

Line 436. Please remove the space before number 35. It should look like this [35]

Line 437. The reference number 351 appears. This is an error that should be checked

Line 448. Please remove one of the dots after "training.”

 Answer: All changes were done as requested

Without a doubt the discussion is the best part of this article, without the previous ones not being good as well. It makes an excellent comparison with the results of previous work which gives this work enormous potential to be revered in the future.

 Answer: Thank you for the positive comment of the discussion

References

Please check each reference one by one and in detail. There are minor formatting errors that require your attention to finally bring you up to speed with a journal like this.

Thanks

Answer: All references were checked

Reviewer #2

Introduction

Line 55. We would appreciate, please, that you include the work of Fierro and Urzua, 2013, in your references: Almonacid Fierro, MA and Urzua Alul, LA. The impact of the supply of creatine monohydrate in canoeing athletes. Revista Iberoamericana de Ciencias de la Actividad Física y el Deporte 2013,1(1):1-19.

Line 70. The inclusion of all football participants is interesting. Referees are part of the game and the following study provides relevant information about this research which we encourage you to consider. Rebolé, M., Castillo, D., Cámara, J., and Yanci, J. “Relationship between the cardiovascular capacity and repeated sprints ability in high-standard soccer referees. Revista Iberoamericana de Ciencias de la Actividad Física y el Deporte 2016,5 (3):49-64.

Line 104. Please remove the space before number 37. It should look like this [37,38]

Line 106. Please, change 2 - 2.5mmol/l by 2-2.5mmol/l

Answer: All changes were done as requested

The introduction is very well written and reviews in depth the state of the art in relation to what is to be researched. However, the identification of the objectives should be better explained at the end of this section.

Answer: Thank you for your valuable observations but we decided not to shorten the admission. We believe that a detailed explanation of the biochemical processes taking place with the participation of creatine significantly facilitates further understanding of the advisability of supplementation with magnesium creatine chelate. However, if the reviewer, after reading the text in the changed form, again decides that the introduction should be shortened, we will do it.

It is implicitly suggested that it is this: "It thus seems fully justified to evaluate the benefits of Cr intake on repeated sprint ability in elite soccer players during 16 weeks of the competitive period". But it leaves serious doubts if what you want to check are the effects of Creatine intake during 16 weeks or its control in serum ... or both. My advice is to make the small adjustments in the text that will allow you to identify without doubt the objective(s). ...The work is excellent!

The main objective was to evaluate the long term effects of Cr supplementation on repeated sprint abilities, crucial for soccer without the classical loading phase, which often causes gastrointestinal problems or muscle cramping. We incorporated the intervention period into the 16 weeks second round competition to obtain information about fitness variables, predominant speed endurance, speed as well as changes in body mass and chosen biochemical variables related to buffering capacity.

Answer: Thank you for your valuable attention. We have added the above text to the introduction.

Material and Methods

Line 112. Please include a dot after the parenthesis (PG= 10).

Line 123. Please include the ethics committee approval code.

Line 127. Please include which 24-hour validated test you have used and include the reference.

Line 136. Please remove one of the dots after "sessions”.

Line 141. Please separate the letter "l" from 1.5l

Line 182. I congratulate you and thank you for informing us about the effect-size

Answer: All changes were done as requested

Results

Line 241. Please remove one of the dots after "intervention. .”- done

Line 241. In the last two rows of the table, the comma (,) has been used as a decimal separator. Please replace all commas with dots.- done

Line 243. Figure 3 > In Figure 3, the variable pH rest does not show data for PG and the variable LA rest does not show values for any of the groups. This must be solved or explained because it confuses the reader.

Line 245. In Figure 3, as you have done in previous tables and figures, include the description of the abbreviations that appear.

 Answer: Thank you for valuable comments. Finally figure 3 was deleted.

Discussion

Line 436. Please remove the space before number 35. It should look like this [35]

Line 437. The reference number 351 appears. This is an error that should be checked

Line 448. Please remove one of the dots after "training.”

 Answer: All changes were done as requested

Without a doubt the discussion is the best part of this article, without the previous ones not being good as well. It makes an excellent comparison with the results of previous work which gives this work enormous potential to be revered in the future.

 Answer: Thank you for the positive comment of the discussion

References

Please check each reference one by one and in detail. There are minor formatting errors that require your attention to finally bring you up to speed with a journal like this.

Thanks

Answer: All references were checked

Reviewer #2

Introduction

Line 55. We would appreciate, please, that you include the work of Fierro and Urzua, 2013, in your references: Almonacid Fierro, MA and Urzua Alul, LA. The impact of the supply of creatine monohydrate in canoeing athletes. Revista Iberoamericana de Ciencias de la Actividad Física y el Deporte 2013,1(1):1-19.

Line 70. The inclusion of all football participants is interesting. Referees are part of the game and the following study provides relevant information about this research which we encourage you to consider. Rebolé, M., Castillo, D., Cámara, J., and Yanci, J. “Relationship between the cardiovascular capacity and repeated sprints ability in high-standard soccer referees. Revista Iberoamericana de Ciencias de la Actividad Física y el Deporte 2016,5 (3):49-64.

Line 104. Please remove the space before number 37. It should look like this [37,38]

Line 106. Please, change 2 - 2.5mmol/l by 2-2.5mmol/l

Answer: All changes were done as requested

The introduction is very well written and reviews in depth the state of the art in relation to what is to be researched. However, the identification of the objectives should be better explained at the end of this section.

Answer: Thank you for your valuable observations but we decided not to shorten the admission. We believe that a detailed explanation of the biochemical processes taking place with the participation of creatine significantly facilitates further understanding of the advisability of supplementation with magnesium creatine chelate. However, if the reviewer, after reading the text in the changed form, again decides that the introduction should be shortened, we will do it.

It is implicitly suggested that it is this: "It thus seems fully justified to evaluate the benefits of Cr intake on repeated sprint ability in elite soccer players during 16 weeks of the competitive period". But it leaves serious doubts if what you want to check are the effects of Creatine intake during 16 weeks or its control in serum ... or both. My advice is to make the small adjustments in the text that will allow you to identify without doubt the objective(s). ...The work is excellent!

The main objective was to evaluate the long term effects of Cr supplementation on repeated sprint abilities, crucial for soccer without the classical loading phase, which often causes gastrointestinal problems or muscle cramping. We incorporated the intervention period into the 16 weeks second round competition to obtain information about fitness variables, predominant speed endurance, speed as well as changes in body mass and chosen biochemical variables related to buffering capacity.

Answer: Thank you for your valuable attention. We have added the above text to the introduction.

Material and Methods

Line 112. Please include a dot after the parenthesis (PG= 10).

Line 123. Please include the ethics committee approval code.

Line 127. Please include which 24-hour validated test you have used and include the reference.

Line 136. Please remove one of the dots after "sessions”.

Line 141. Please separate the letter "l" from 1.5l

Line 182. I congratulate you and thank you for informing us about the effect-size

Answer: All changes were done as requested

Results

Line 241. Please remove one of the dots after "intervention. .”- done

Line 241. In the last two rows of the table, the comma (,) has been used as a decimal separator. Please replace all commas with dots.- done

Line 243. Figure 3 > In Figure 3, the variable pH rest does not show data for PG and the variable LA rest does not show values for any of the groups. This must be solved or explained because it confuses the reader.

Line 245. In Figure 3, as you have done in previous tables and figures, include the description of the abbreviations that appear.

 Answer: Thank you for valuable comments. Finally figure 3 was deleted.

Discussion

Line 436. Please remove the space before number 35. It should look like this [35]

Line 437. The reference number 351 appears. This is an error that should be checked

Line 448. Please remove one of the dots after "training.”

 Answer: All changes were done as requested

Without a doubt the discussion is the best part of this article, without the previous ones not being good as well. It makes an excellent comparison with the results of previous work which gives this work enormous potential to be revered in the future.

 Answer: Thank you for the positive comment of the discussion

References

Please check each reference one by one and in detail. There are minor formatting errors that require your attention to finally bring you up to speed with a journal like this.

Thanks

Answer: All references were checked

Reviewer #2

Introduction

Line 55. We would appreciate, please, that you include the work of Fierro and Urzua, 2013, in your references: Almonacid Fierro, MA and Urzua Alul, LA. The impact of the supply of creatine monohydrate in canoeing athletes. Revista Iberoamericana de Ciencias de la Actividad Física y el Deporte 2013,1(1):1-19.

Line 70. The inclusion of all football participants is interesting. Referees are part of the game and the following study provides relevant information about this research which we encourage you to consider. Rebolé, M., Castillo, D., Cámara, J., and Yanci, J. “Relationship between the cardiovascular capacity and repeated sprints ability in high-standard soccer referees. Revista Iberoamericana de Ciencias de la Actividad Física y el Deporte 2016,5 (3):49-64.

Line 104. Please remove the space before number 37. It should look like this [37,38]

Line 106. Please, change 2 - 2.5mmol/l by 2-2.5mmol/l

Answer: All changes were done as requested

The introduction is very well written and reviews in depth the state of the art in relation to what is to be researched. However, the identification of the objectives should be better explained at the end of this section.

Answer: Thank you for your valuable observations but we decided not to shorten the admission. We believe that a detailed explanation of the biochemical processes taking place with the participation of creatine significantly facilitates further understanding of the advisability of supplementation with magnesium creatine chelate. However, if the reviewer, after reading the text in the changed form, again decides that the introduction should be shortened, we will do it.

It is implicitly suggested that it is this: "It thus seems fully justified to evaluate the benefits of Cr intake on repeated sprint ability in elite soccer players during 16 weeks of the competitive period". But it leaves serious doubts if what you want to check are the effects of Creatine intake during 16 weeks or its control in serum ... or both. My advice is to make the small adjustments in the text that will allow you to identify without doubt the objective(s). ...The work is excellent!

The main objective was to evaluate the long term effects of Cr supplementation on repeated sprint abilities, crucial for soccer without the classical loading phase, which often causes gastrointestinal problems or muscle cramping. We incorporated the intervention period into the 16 weeks second round competition to obtain information about fitness variables, predominant speed endurance, speed as well as changes in body mass and chosen biochemical variables related to buffering capacity.

Answer: Thank you for your valuable attention. We have added the above text to the introduction.

Material and Methods

Line 112. Please include a dot after the parenthesis (PG= 10).

Line 123. Please include the ethics committee approval code.

Line 127. Please include which 24-hour validated test you have used and include the reference.

Line 136. Please remove one of the dots after "sessions”.

Line 141. Please separate the letter "l" from 1.5l

Line 182. I congratulate you and thank you for informing us about the effect-size

Answer: All changes were done as requested

Results

Line 241. Please remove one of the dots after "intervention. .”- done

Line 241. In the last two rows of the table, the comma (,) has been used as a decimal separator. Please replace all commas with dots.- done

Line 243. Figure 3 > In Figure 3, the variable pH rest does not show data for PG and the variable LA rest does not show values for any of the groups. This must be solved or explained because it confuses the reader.

Line 245. In Figure 3, as you have done in previous tables and figures, include the description of the abbreviations that appear.

 Answer: Thank you for valuable comments. Finally figure 3 was deleted.

Discussion

Line 436. Please remove the space before number 35. It should look like this [35]

Line 437. The reference number 351 appears. This is an error that should be checked

Line 448. Please remove one of the dots after "training.”

 Answer: All changes were done as requested

Without a doubt the discussion is the best part of this article, without the previous ones not being good as well. It makes an excellent comparison with the results of previous work which gives this work enormous potential to be revered in the future.

 Answer: Thank you for the positive comment of the discussion

References

Please check each reference one by one and in detail. There are minor formatting errors that require your attention to finally bring you up to speed with a journal like this.

Thanks

Answer: All references were checked

Reviewer #2

Introduction

Line 55. We would appreciate, please, that you include the work of Fierro and Urzua, 2013, in your references: Almonacid Fierro, MA and Urzua Alul, LA. The impact of the supply of creatine monohydrate in canoeing athletes. Revista Iberoamericana de Ciencias de la Actividad Física y el Deporte 2013,1(1):1-19.

Line 70. The inclusion of all football participants is interesting. Referees are part of the game and the following study provides relevant information about this research which we encourage you to consider. Rebolé, M., Castillo, D., Cámara, J., and Yanci, J. “Relationship between the cardiovascular capacity and repeated sprints ability in high-standard soccer referees. Revista Iberoamericana de Ciencias de la Actividad Física y el Deporte 2016,5 (3):49-64.

Line 104. Please remove the space before number 37. It should look like this [37,38]

Line 106. Please, change 2 - 2.5mmol/l by 2-2.5mmol/l

Answer: All changes were done as requested

The introduction is very well written and reviews in depth the state of the art in relation to what is to be researched. However, the identification of the objectives should be better explained at the end of this section.

Answer: Thank you for your valuable observations but we decided not to shorten the admission. We believe that a detailed explanation of the biochemical processes taking place with the participation of creatine significantly facilitates further understanding of the advisability of supplementation with magnesium creatine chelate. However, if the reviewer, after reading the text in the changed form, again decides that the introduction should be shortened, we will do it.

It is implicitly suggested that it is this: "It thus seems fully justified to evaluate the benefits of Cr intake on repeated sprint ability in elite soccer players during 16 weeks of the competitive period". But it leaves serious doubts if what you want to check are the effects of Creatine intake during 16 weeks or its control in serum ... or both. My advice is to make the small adjustments in the text that will allow you to identify without doubt the objective(s). ...The work is excellent!

The main objective was to evaluate the long term effects of Cr supplementation on repeated sprint abilities, crucial for soccer without the classical loading phase, which often causes gastrointestinal problems or muscle cramping. We incorporated the intervention period into the 16 weeks second round competition to obtain information about fitness variables, predominant speed endurance, speed as well as changes in body mass and chosen biochemical variables related to buffering capacity.

Answer: Thank you for your valuable attention. We have added the above text to the introduction.

Material and Methods

Line 112. Please include a dot after the parenthesis (PG= 10).

Line 123. Please include the ethics committee approval code.

Line 127. Please include which 24-hour validated test you have used and include the reference.

Line 136. Please remove one of the dots after "sessions”.

Line 141. Please separate the letter "l" from 1.5l

Line 182. I congratulate you and thank you for informing us about the effect-size

Answer: All changes were done as requested

Results

Line 241. Please remove one of the dots after "intervention. .”- done

Line 241. In the last two rows of the table, the comma (,) has been used as a decimal separator. Please replace all commas with dots.- done

Line 243. Figure 3 > In Figure 3, the variable pH rest does not show data for PG and the variable LA rest does not show values for any of the groups. This must be solved or explained because it confuses the reader.

Line 245. In Figure 3, as you have done in previous tables and figures, include the description of the abbreviations that appear.

 Answer: Thank you for valuable comments. Finally figure 3 was deleted.

Discussion

Line 436. Please remove the space before number 35. It should look like this [35]

Line 437. The reference number 351 appears. This is an error that should be checked

Line 448. Please remove one of the dots after "training.”

 Answer: All changes were done as requested

Without a doubt the discussion is the best part of this article, without the previous ones not being good as well. It makes an excellent comparison with the results of previous work which gives this work enormous potential to be revered in the future.

 Answer: Thank you for the positive comment of the discussion

References

Please check each reference one by one and in detail. There are minor formatting errors that require your attention to finally bring you up to speed with a journal like this.

Thanks

Answer: All references were checked

Reviewer #2

Introduction

Line 55. We would appreciate, please, that you include the work of Fierro and Urzua, 2013, in your references: Almonacid Fierro, MA and Urzua Alul, LA. The impact of the supply of creatine monohydrate in canoeing athletes. Revista Iberoamericana de Ciencias de la Actividad Física y el Deporte 2013,1(1):1-19.

Line 70. The inclusion of all football participants is interesting. Referees are part of the game and the following study provides relevant information about this research which we encourage you to consider. Rebolé, M., Castillo, D., Cámara, J., and Yanci, J. “Relationship between the cardiovascular capacity and repeated sprints ability in high-standard soccer referees. Revista Iberoamericana de Ciencias de la Actividad Física y el Deporte 2016,5 (3):49-64.

Line 104. Please remove the space before number 37. It should look like this [37,38]

Line 106. Please, change 2 - 2.5mmol/l by 2-2.5mmol/l

Answer: All changes were done as requested

The introduction is very well written and reviews in depth the state of the art in relation to what is to be researched. However, the identification of the objectives should be better explained at the end of this section.

Answer: Thank you for your valuable observations but we decided not to shorten the admission. We believe that a detailed explanation of the biochemical processes taking place with the participation of creatine significantly facilitates further understanding of the advisability of supplementation with magnesium creatine chelate. However, if the reviewer, after reading the text in the changed form, again decides that the introduction should be shortened, we will do it.

It is implicitly suggested that it is this: "It thus seems fully justified to evaluate the benefits of Cr intake on repeated sprint ability in elite soccer players during 16 weeks of the competitive period". But it leaves serious doubts if what you want to check are the effects of Creatine intake during 16 weeks or its control in serum ... or both. My advice is to make the small adjustments in the text that will allow you to identify without doubt the objective(s). ...The work is excellent!

The main objective was to evaluate the long term effects of Cr supplementation on repeated sprint abilities, crucial for soccer without the classical loading phase, which often causes gastrointestinal problems or muscle cramping. We incorporated the intervention period into the 16 weeks second round competition to obtain information about fitness variables, predominant speed endurance, speed as well as changes in body mass and chosen biochemical variables related to buffering capacity.

Answer: Thank you for your valuable attention. We have added the above text to the introduction.

Material and Methods

Line 112. Please include a dot after the parenthesis (PG= 10).

Line 123. Please include the ethics committee approval code.

Line 127. Please include which 24-hour validated test you have used and include the reference.

Line 136. Please remove one of the dots after "sessions”.

Line 141. Please separate the letter "l" from 1.5l

Line 182. I congratulate you and thank you for informing us about the effect-size

Answer: All changes were done as requested

Results

Line 241. Please remove one of the dots after "intervention. .”- done

Line 241. In the last two rows of the table, the comma (,) has been used as a decimal separator. Please replace all commas with dots.- done

Line 243. Figure 3 > In Figure 3, the variable pH rest does not show data for PG and the variable LA rest does not show values for any of the groups. This must be solved or explained because it confuses the reader.

Line 245. In Figure 3, as you have done in previous tables and figures, include the description of the abbreviations that appear.

 Answer: Thank you for valuable comments. Finally figure 3 was deleted.

Discussion

Line 436. Please remove the space before number 35. It should look like this [35]

Line 437. The reference number 351 appears. This is an error that should be checked

Line 448. Please remove one of the dots after "training.”

 Answer: All changes were done as requested

Without a doubt the discussion is the best part of this article, without the previous ones not being good as well. It makes an excellent comparison with the results of previous work which gives this work enormous potential to be revered in the future.

 Answer: Thank you for the positive comment of the discussion

References

Please check each reference one by one and in detail. There are minor formatting errors that require your attention to finally bring you up to speed with a journal like this.

Thanks

Answer: All references were checked

Reviewer #2

Introduction

Line 55. We would appreciate, please, that you include the work of Fierro and Urzua, 2013, in your references: Almonacid Fierro, MA and Urzua Alul, LA. The impact of the supply of creatine monohydrate in canoeing athletes. Revista Iberoamericana de Ciencias de la Actividad Física y el Deporte 2013,1(1):1-19.

Line 70. The inclusion of all football participants is interesting. Referees are part of the game and the following study provides relevant information about this research which we encourage you to consider. Rebolé, M., Castillo, D., Cámara, J., and Yanci, J. “Relationship between the cardiovascular capacity and repeated sprints ability in high-standard soccer referees. Revista Iberoamericana de Ciencias de la Actividad Física y el Deporte 2016,5 (3):49-64.

Line 104. Please remove the space before number 37. It should look like this [37,38]

Line 106. Please, change 2 - 2.5mmol/l by 2-2.5mmol/l

Answer: All changes were done as requested

The introduction is very well written and reviews in depth the state of the art in relation to what is to be researched. However, the identification of the objectives should be better explained at the end of this section.

Answer: Thank you for your valuable observations but we decided not to shorten the admission. We believe that a detailed explanation of the biochemical processes taking place with the participation of creatine significantly facilitates further understanding of the advisability of supplementation with magnesium creatine chelate. However, if the reviewer, after reading the text in the changed form, again decides that the introduction should be shortened, we will do it.

It is implicitly suggested that it is this: "It thus seems fully justified to evaluate the benefits of Cr intake on repeated sprint ability in elite soccer players during 16 weeks of the competitive period". But it leaves serious doubts if what you want to check are the effects of Creatine intake during 16 weeks or its control in serum ... or both. My advice is to make the small adjustments in the text that will allow you to identify without doubt the objective(s). ...The work is excellent!

The main objective was to evaluate the long term effects of Cr supplementation on repeated sprint abilities, crucial for soccer without the classical loading phase, which often causes gastrointestinal problems or muscle cramping. We incorporated the intervention period into the 16 weeks second round competition to obtain information about fitness variables, predominant speed endurance, speed as well as changes in body mass and chosen biochemical variables related to buffering capacity.

Answer: Thank you for your valuable attention. We have added the above text to the introduction.

Material and Methods

Line 112. Please include a dot after the parenthesis (PG= 10).

Line 123. Please include the ethics committee approval code.

Line 127. Please include which 24-hour validated test you have used and include the reference.

Line 136. Please remove one of the dots after "sessions”.

Line 141. Please separate the letter "l" from 1.5l

Line 182. I congratulate you and thank you for informing us about the effect-size

Answer: All changes were done as requested

Results

Line 241. Please remove one of the dots after "intervention. .”- done

Line 241. In the last two rows of the table, the comma (,) has been used as a decimal separator. Please replace all commas with dots.- done

Line 243. Figure 3 > In Figure 3, the variable pH rest does not show data for PG and the variable LA rest does not show values for any of the groups. This must be solved or explained because it confuses the reader.

Line 245. In Figure 3, as you have done in previous tables and figures, include the description of the abbreviations that appear.

 Answer: Thank you for valuable comments. Finally figure 3 was deleted.

Discussion

Line 436. Please remove the space before number 35. It should look like this [35]

Line 437. The reference number 351 appears. This is an error that should be checked

Line 448. Please remove one of the dots after "training.”

 Answer: All changes were done as requested

Without a doubt the discussion is the best part of this article, without the previous ones not being good as well. It makes an excellent comparison with the results of previous work which gives this work enormous potential to be revered in the future.

 Answer: Thank you for the positive comment of the discussion

References

Please check each reference one by one and in detail. There are minor formatting errors that require your attention to finally bring you up to speed with a journal like this.

Thanks

Answer: All references were checked

Reviewer #2

Introduction

Line 55. We would appreciate, please, that you include the work of Fierro and Urzua, 2013, in your references: Almonacid Fierro, MA and Urzua Alul, LA. The impact of the supply of creatine monohydrate in canoeing athletes. Revista Iberoamericana de Ciencias de la Actividad Física y el Deporte 2013,1(1):1-19.

Line 70. The inclusion of all football participants is interesting. Referees are part of the game and the following study provides relevant information about this research which we encourage you to consider. Rebolé, M., Castillo, D., Cámara, J., and Yanci, J. “Relationship between the cardiovascular capacity and repeated sprints ability in high-standard soccer referees. Revista Iberoamericana de Ciencias de la Actividad Física y el Deporte 2016,5 (3):49-64.

Line 104. Please remove the space before number 37. It should look like this [37,38]

Line 106. Please, change 2 - 2.5mmol/l by 2-2.5mmol/l

Answer: All changes were done as requested

The introduction is very well written and reviews in depth the state of the art in relation to what is to be researched. However, the identification of the objectives should be better explained at the end of this section.

Answer: Thank you for your valuable observations but we decided not to shorten the admission. We believe that a detailed explanation of the biochemical processes taking place with the participation of creatine significantly facilitates further understanding of the advisability of supplementation with magnesium creatine chelate. However, if the reviewer, after reading the text in the changed form, again decides that the introduction should be shortened, we will do it.

It is implicitly suggested that it is this: "It thus seems fully justified to evaluate the benefits of Cr intake on repeated sprint ability in elite soccer players during 16 weeks of the competitive period". But it leaves serious doubts if what you want to check are the effects of Creatine intake during 16 weeks or its control in serum ... or both. My advice is to make the small adjustments in the text that will allow you to identify without doubt the objective(s). ...The work is excellent!

The main objective was to evaluate the long term effects of Cr supplementation on repeated sprint abilities, crucial for soccer without the classical loading phase, which often causes gastrointestinal problems or muscle cramping. We incorporated the intervention period into the 16 weeks second round competition to obtain information about fitness variables, predominant speed endurance, speed as well as changes in body mass and chosen biochemical variables related to buffering capacity.

Answer: Thank you for your valuable attention. We have added the above text to the introduction.

Material and Methods

Line 112. Please include a dot after the parenthesis (PG= 10).

Line 123. Please include the ethics committee approval code.

Line 127. Please include which 24-hour validated test you have used and include the reference.

Line 136. Please remove one of the dots after "sessions”.

Line 141. Please separate the letter "l" from 1.5l

Line 182. I congratulate you and thank you for informing us about the effect-size

Answer: All changes were done as requested

Results

Line 241. Please remove one of the dots after "intervention. .”- done

Line 241. In the last two rows of the table, the comma (,) has been used as a decimal separator. Please replace all commas with dots.- done

Line 243. Figure 3 > In Figure 3, the variable pH rest does not show data for PG and the variable LA rest does not show values for any of the groups. This must be solved or explained because it confuses the reader.

Line 245. In Figure 3, as you have done in previous tables and figures, include the description of the abbreviations that appear.

 Answer: Thank you for valuable comments. Finally figure 3 was deleted.

Discussion

Line 436. Please remove the space before number 35. It should look like this [35]

Line 437. The reference number 351 appears. This is an error that should be checked

Line 448. Please remove one of the dots after "training.”

 Answer: All changes were done as requested

Without a doubt the discussion is the best part of this article, without the previous ones not being good as well. It makes an excellent comparison with the results of previous work which gives this work enormous potential to be revered in the future.

 Answer: Thank you for the positive comment of the discussion

References

Please check each reference one by one and in detail. There are minor formatting errors that require your attention to finally bring you up to speed with a journal like this.

Thanks

Answer: All references were checked

Round 2

Reviewer 1 Report

I really like this paper, and the results are important, but it is still just entirely too long. It needs to be shortened, because as it currently is written the authors results (and thus the actual work they did) are being lost to the noise of the textbook material that is not important to explain there results to the reader. I have provided a few example below of things they could easily get rid of but I'm not able to tell them everything.  

I commend the authors for making the changes they did. I still think its important you define what "elite" means. Simply providing the demographics does not tell me it they're an elite athlete. I would state at what level of competition they are playing at, which is a common way to report levels of "elite." 

The discussion is still entirely too long. Discussion typically are about 1500 - 2000 words. The current discussion comes in at ~3000. Statements like the following are not required in an original research article, they are statements you would use in a review paper: 

For over 30 years, numerous studies have been conducted regarding creatine supplementation in competitive athletes [1,2,7,14,45].

For many years, next to caffeine, Cr has been the most commonly used supplement by athletes training speed and strength sport disciplines [18,49].

In addition, Cr supplementation is safe, does not cause gastrointestinal disorders [51-52] and is recommended by many sport nutrition authorities and institutions [2,6,7,35,40,53].

The entire paragraph from 298 to 309 can be removed. This is information for a textbook. 

None of this information is crucial for the reader to understand the results of your study. Its great information but it belongs in a review paper or textbook. Please. These are only a few examples of information that is not required for a reader to interpret your results. There are many more, please condense and make the discussion more concise. 

Reference 54 is not an appropriate reference for improved bioavailability with Magnesium Creatine Chelate compared to monohydrate for many reason. One being in rats, and you compared humans, and two bring not tissue concentrations of creatine were measured thus you can’t established bioavailability.

Author Response

Dear Reviewer

Thank you for your valuable comments. As you requested, we corrected all
comments and shortened the discussion considerably. We also explained the
meaning of the word elite.